# A co-designed website (FindWays) to improve mental health literacy of parents of children with mental health problems: Protocol for a pilot randomised controlled trial

Daniel Peyton[1,2]*, Greg Wadley[3], Naomi Hackworth[4,5,6], Anneke Grobler[2,6], Harriet Hiscock[1,2,7]

1 Health Services and Economics, Murdoch Children's Research Institute, Melbourne, Victoria, Australia, 2 Department of Paediatrics, University of Melbourne, Melbourne, Victoria, Australia, 3 School of Computing and Information Systems, The University of Melbourne, Melbourne, Australia, 4 Judith Lumley Centre, La Trobe University, Melbourne, Australia, 5 Parenting Research Centre, East Melbourne, Australia, 6 Murdoch Children's Research Institute, Parkville, Australia, 7 Health Services Research Unit, The Royal Children's Hospital, Melbourne, Victoria, Australia

* daniel.peyton@mcri.edu.au

**Data Availability Statement:** The data are not expected to be made available at an individual level. The researchers do not have ethics consent from

## Abstract

### Background

Mental health problems, such as behavioural and emotional problems, are prevalent in children. These problems can have long lasting, detrimental effects on the child, their parents and society. Most children with a mental health problem do not receive professional help. Those that do get help can face long wait times. While waiting, parents want to learn how they can help their child. To address this need, we co-designed a new website to help parents find ways of helping their child's mental health problem while waiting to get specialist help.

### Objectives

To assess the acceptability and feasibility of a new co-designed website, FindWays, through a pilot randomised controlled trial. The protocol is registered with ISRCTN (ISRCTN64605513).

### Methods

This study will recruit up to 60 parents of children aged two-twelve years old referred to a paediatrician for behavioural and/or emotional problems. Participants will be randomly allocated by computer generated number sequence to either the intervention or control group. Intervention group participants will receive access to the FindWays website to help them manage their child's mental health problem while they wait to see the paediatrician. Acceptability and feasibility will be assessed over the 4-month intervention through mixed methods including: recruitment, adherence, retention, net promoter score (quantitative measures) and semi-structured interviews to gain an in-depth understanding of parents' experience

participants to share individual-level data with other organisations. Data will be stored by the Murdoch Children's Research Institute. Data are available from The Royal Children's Hospital, Melbourne Human Research Ethics Committee (contact via +61 3 9345 5044) for researchers who meet the criteria for access to confidential data.

**Funding:** The authors would like to thank the Murdoch Children's Research Institute (https://www.mcri.edu.au/research/research-areas/population-health) and Charity Drive Days (https://charitydrivedays.com.au/) for funding support in designing and building the FindWays website. DP is supported by the National Health and Medical Research Council Postgraduate Scholarship (APP1189935). HH is supported by NHMRC Practitioner Fellowship Award (APP1136222). https://www.nhmrc.gov.au/ The Victorian Government's Operational Infrastructure Support Program supported research at the Murdoch Children's Research Institute. The funders had no role in study design, data collection and analysis, decision to publish, or preparation of the manuscript.

**Competing interests:** The authors have declared that no competing interests exist.

and potential adverse effects (qualitative measure). Secondary outcomes measured by parent survey at 4-months post randomisation include child mental health, parent mental health, impact of the child's mental health problem on their functioning and family, and health service use and associated costs.

## Results

Recruitment commenced June 2022 with publication expected in October 2023.

## Conclusion

This study will provide novel data on the acceptability and feasibility of a new website co-designed with parents to help them find ways of managing their child's behaviour and emotions.

## Introduction

### Background

**Mental health problems and their long term effects.**    Diagnosed mental health disorders, such as disruptive behaviour, anxiety and mood problems, affect one in seven Australian children [1]. However, mental health problems, which can still impact the child but may not reach diagnostic criteria for a mental health disorder. These more common behavioural and emotional problems likely affect around one in four children [1]. These common problems impact children's short and long term outcomes including difficulty with relationships [2], reduced educational achievement [2], increased risk of criminality [3, 4], alcohol and drug problems [5] and lower earning potential [3].

Half of all adult mental health disorders start in childhood [6]. Evidence-based treatments including parenting interventions have been shown to improve child behavioural and emotional problems, including face to face therapy, group parenting programs (e.g., Tuning in to Kids) and online parenting interventions (e.g., computer assisted cognitive behaviour therapy, such as BRAVE online) [7, 8]. Often, interventions target the parents given that parenting is a key modifiable factor in child mental health [9, 10].

Despite the availability of a range of evidence-based interventions for mental health problems, most children with a mental health disorder do not receive professional help [11–14]. Reasons for this are multifactorial and include accessibility barriers, stigma, and poor parental (and professional) mental health literacy [11]. Mental health literacy is the "knowledge and beliefs about mental disorders which aid their recognition, management or prevention" [15][p182]. This can include knowing where a child can go to get mental health help, or strategies parents can implement to help alleviate their child's problem.

Qualitative studies have shown that parents often face difficulty navigating the maze of services for mental health [16, 17]. Parents can find it hard to know where to go or who can help. Many will end up seeing a paediatrician to help their child [18].

In Australia, the paediatrician is the largest provider of longitudinal care of children with a mental health disorder [11]. Paediatricians are often an 'early port of call' for children with mental health disorders and play a crucial role in diagnosis, exclusion of medical causes of child mental health problems and prescription of medication. Paediatricians generally do not deliver behaviour management therapy nor are they trained in cognitive behavioural therapy. They generally refer to other providers for these interventions.

With few publicly funded paediatric services available, anecdotal reports suggest wait times in excess of six to twelve months to access a paediatrician in Australia. Pre-COVID (2019), mean wait time to access a paediatrician in the states of Victoria and South Australia was 44 days with four out of 10 providers closing their books to new patients mid-year [19]. This problem of accessibility has worsened due to the COVID-19 pandemic, especially for families who cannot afford private care.

**Parents want support and information while they wait to see a mental health specialist.** Parents want information while they wait to access mental health care [20]. This includes a better understanding of how the mental health system works and how to access other forms of treatment to help their child while they wait, such as group therapy, online interventions or telephone support [20].

There is little qualitative data on parent's information preferences while waiting to see a specialist mental health professional. From first author's qualitative interviews with 16 parents waiting to get help from a paediatrician (unpublished data), inductive content analysis revealed that parents want to understand whether their child's problem is normal or not, what they could try while they waited to see the paediatrician, which services could help their child's problem, and a list of available professional help in their community.

**Why a digital health intervention might help.** Digital health interventions (DHIs) may be able to address these information gaps identified by parents. DHIs are desirable for the following reasons: they facilitate the rapid exchange of tailored information to consumers, are scalable, are liked by parents, can have vast reach, are accessible on demand, and generally thought to be cost-effective [21].

Our recent systematic review identified only five studies examining the effects of a DHI on parents' mental health literacy or help-seeking for their child's mental health problem [22]. There is some evidence that DHIs, such as websites and emailed PowerPoint presentations, may help improve mental health literacy [23–25]. However, the review included only one randomised controlled trial [26]. A Finnish study measured help-seeking behaviours of parents of four year old children with a recognised behaviour problem at a routine four year old check with a universal service. The randomised controlled trial found an information website, coupled with a telephone coach decreased uptake of services. However, this decrease in uptake of services was associated with improved child behaviour using a validated measure. This study showed a DHI, with a telephone coach, can affect help-seeking [26]. However, no quantitative studies have shown DHIs can improve uptake of services for children with a mental health problem.

**A new co-designed website to improve help-seeking and child mental health outcomes.** To design a new website to improve parents' knowledge of treatments and reduce barriers to accessing evidence-based therapies for their child, we followed a framework proposed by Gemert-Pijen et al. [27]. This is a user-centred, iterative approach that refines the DHI to match the needs of the user, during every phase of the development cycle [27]. The website was developed in collaboration with parents and clinicians. As part of this process, we undertook a contextual enquiry with parents, three co-design workshops and six one-on-one usability studies. Through this process, we designed and developed a website with the content detailed in the intervention section below.

**Evaluating the feasibility and acceptability of the DHI.** The World Health Organisation recommends evaluating the feasibility of a new DHI, after usability testing and prior to efficacy and effectiveness evaluation [28]. The measurement of feasibility is complicated by the absence of a singular definition of feasibility testing, and the synonymous use of feasibility studies and pilot studies [29, 30]. Feasibility trials typically involve 20–100 participants, and evaluate the suitability of outcome measures, participant acceptance of the intervention (recruitment and retention rates) and DHI usage (e.g. adherence) [28–32].

From our systematic review of five included studies, we found single cohort trials limited the strength of evidence of effectiveness as this methodology is inherently prone to confounding bias [33]. None of the included studies in the systematic review identified or adjusted for the presence of confounding. To address this, we plan to conduct a randomised controlled pilot study with a "routine care" control group. This also allows us to describe differences in efficacy of secondary outcomes between the two groups, accounting for confounding and natural history of child mental health problems over time.

### Aim

We aim to assess the acceptability and feasibility of a co-designed website, FindWays, through a pilot randomised controlled trial using one-to-one allocation in parallel arms. The study will also assess efficacy for outcomes related to child mental health, parent mental health, family impact and health service use and costs.

**Hypothesis.** We hypothesise that FindWays is an acceptable and feasible DHI with the potential to help parents find ways to improve their child's mental health problems while they wait to see a paediatrician.

## Methods

### Trial design

We will conduct a pilot randomised controlled trial with participants randomly assigned to receive either the FindWays website (intervention group) or routine care (control group). We will use a mixed methods evaluation to determine acceptability and feasibility.

### Setting

Participants will be recruited from three general paediatric private outpatient clinics in Geelong, Australia. Geelong is the largest regional city in the state of Victoria. 51,804 children under the age of 15 live in Geelong [34]. At the start of recruitment, there were no public general paediatric outpatient clinics. The list of recruitment sites is available from the website https://tiny.one/findways.

### Eligibility

Participants in the study must be a parent of a child referred by a general practitioner (family doctor) to a paediatrician at one of three private paediatric clinics, for a first appointment to evaluate or manage their child's behavioural or emotional problem. Participants will be assigned to a randomised trial intervention only if they meet all the inclusion criteria and none of the exclusion criteria. Only one parent of a child will be enrolled.

**Inclusion criteria.** Each participant must meet all the following criteria to be enrolled in this trial.

- is a parent or carer of a child between two and 12 years old at the time of randomisation;

- has been referred for a first appointment with a paediatrician to manage a child's behavioural or emotional problem;

- child has a behavioural or emotional problem as listed for review on the referral letter from the child's general practitioner; and

- provides verbal consent that is signed and dated by the researcher (DP).

**Exclusion criteria.** Participants meeting any of the following criteria will be excluded from the trial:

- child has a parent-reported diagnosis of an Intellectual Disability or Autism Spectrum Disorder;

- child is in out of home care; or

- paediatrician decides the child needs an early review (such as for a severe mental health problem or to exclude a medical problem).

## Intervention

**Purpose.** The intervention consists of a new, co-designed website. This website, FindWays, offers parents relevant and specific information on behavioural and emotional problems. This information focuses on ways parents can manage a behavioural or emotional problem.

**Theory.** Human-centred and participatory design methods were used to inform the website's design, as was persuasive technology [35]. These methods align with those suggested by existing theoretical framework for DHIs [27]. In order to encourage behaviour change, the UK Behavioural Insights Team EAST behavioural insights were applied to the design and content of website [36]. The behavioural insights team recommend four principles to encourage a behaviour: make is easy, attractive, social and timely [36].

**Contextual enquiry.** The content of the website was informed by inductive content analysis of 16 qualitative interviews completed in early 2020 by DP with parents recently referred to a paediatrician to manage their child's behavioural or emotional problem.

**Prototyping.** We conducted a further three co-design sessions in 2022 with five parents to inform the functional requirements and look and feel of the website. We designed the intervention using an iterative process, sketching wireframes and mock-ups of the website using free sketch on pen and paper, PowerPoint, Adobe XD and finally in Figma, and asking parents for feedback on content and usability. The development of the prototype in Figma and the website in Webflow was assisted by digital technology design company *Curve Tomorrow* [37]. The prototype was built in Webflow utilising the Content Management System. Usability testing was completed with a further six parents. These usability tests were completed as one-on-one interviews, either face-to-face in a paediatric clinic or over Zoom.

**Technical.** This static website contains 60 pages of content, 47 videos and a list of 63 local and online services for child mental health, hosted on Webflow. The parent can select specific ages (preschool or primary school aged) and problems (15 in total) to find tailored information relevant to their child.

**Content.** The website contains the following elements: i) strategies for parents to try at home based on a program logic that identifies evidence-based strategies associated with positive child mental health outcomes, ii) descriptions of the roles of professionals and programs that can help a child's mental health problem, and iii) lists of local professionals and available programs. The website content includes videos, written content, and links to relevant external content and services, including local psychologists, occupational therapists and evidence-based programs. The content is tailored to the child's age and specific sub-problem (e.g., separation anxiety, tantrums), as identified by the parent. The parent can choose what help they want for their child, but the website guides parents, who self-identify through the website that they want more help, to a local professional or program that can help their child's specific sub-problem.

The parenting strategies provided by the website reference existing information sources such as the Raising Children Network, Beyond Blue and the Diagnostic Standard Manual-V (DSM-V) [38]. The Raising Children Network and Beyond Blue are both Australian websites

providing mental health content for consumers, funded in part by the Australian Government. DSM-V is a diagnostic tool, published by the American Psychiatric Association (APA) and used by clinicians as a standardised guide to classify mental health disorders [38].

All content on FindWays was written by first author DP, a general paediatrician. The content underwent a quality review process including review by another paediatrician (HH). Video scripts were reviewed and approved by the paediatricians in Geelong. The content was also reviewed by NH (a psychologist and Director, Raising Children Network) to ensure the tone is appropriate for families.

**Prompts.** Throughout the intervention, parents will receive scheduled prompts to use the website. These prompts, delivered by SMS and email, will contain static information drawn from the website, and a link to the website. These will be delivered at regular, pre-scheduled intervals and are not tailored. Parents will be able to request the prompts stop at any time, either by email or text message.

**Control group.** The control group will receive routine care. This care is provided by usual providers (e.g., GP, existing online resources, teachers at school). The control group will not be provided access to the FindWays website. The FindWays website is not listed on search engines, and the URL is not publicly available.

## Outcomes

Outcomes of this trial will be measured using mixed methods. Data will be collected at baseline and at four months post randomisation, except for DHI usage data which will be measured continuously by Google Analytics and downloaded manually every month during the trial. See Table 1 for a summary of the outcome measures and data collection time points.

**Primary outcomes measures.** Despite the lack of a singular definition of feasibility, they typically involve measures of recruitment rates, attrition rates, evidence of harm, user satisfaction and DHI usage [28, 29] and will be assessed in the following ways:

- feasibility will be measured by recruitment rates including the proportion of potentially eligible parents who consent to participate in the study and complete outcome measures;

- adherence will be measured by the number of consenting participants in the intervention group who access the website as measured by Google Analytics;

- At 4-months post randomisation:

  ○ acceptability measured by likelihood of recommending the intervention measured by Likert scale Net Promoter Score

  ○ acceptability measured by usability and safety feedback measured by semi-structured phone interview

  ○ retention measured by number of consented parents who complete all baseline measures and final measures

  ○ adherence measured by tasks completed on the website (e.g., strategies found, suitable providers found) measured by parent survey.

**Secondary outcome measures.** Secondary outcomes measured by parent-reported surveys at baseline and 4-months post randomisation include:

- Child behaviour and emotions measured by the Strengths and Difficulties Questionnaire (SDQ)—a 25-item, validated measure of psychological attributes for 2–17 year olds, as reported by the parents [39].

**Table 1. Study objectives, data sources, data collection time points and outcome measures.**

| Demographic data, Primary and Secondary Objectives | Data Sources | Method of collection | Data Collection Time points | Outcomes |
|---|---|---|---|---|
| **Primary outcomes** | | | | |
| **Feasibility–recruitment and retention** | Clinic database | Print copy of database | Recorded at weekly intervals for the duration of recruitment | How many eligible patients were flagged by the paediatrician, how many were contacted by the clinic, and how many agreed to pass on their details to the researcher and how many were randomised to the intervention/control, and completed their survey at four months. |
| **Adherence and usage (intervention group only)** | Google Analytics | Data extracted manually from Google Analytics | Baseline and Weekly intervals for 4 months | Number of page views, individual sessions, time spent on each page for each participant. |
| | | | | Strategies viewed, provider profiles viewed, programs viewed. |
| **Acceptability–usability and safety (intervention group only)** | Parent qualitative Interview | Qualitative interview with a subsample of approximately 20 parents from Intervention group | 4 months | Understanding of the parent's experience of using the platform, facilitators, and barriers to use, and any perceived adverse outcomes |
| **Adherence–task completion (intervention group only)** | Parent Survey | Online via REDCap | 4 months | Whether parents reported viewing any of the resources within the website, and whether this changed the parent's behaviour (e.g., help-seeking, implementing new strategies) |
| **Acceptability— recommendation (intervention group only)** | Parent Survey | Online via REDCap | 4 months | Net Promoter Score measured via 10-point Likert scale. |
| **Secondary outcomes** | | | | |
| **Secondary outcome: child behaviour and emotions** | Parent Survey | Control and intervention group surveys completed online via REDCap. | Baseline and 4 months | Strengths and Difficulties Questionnaire[a] (SDQ) |
| **Family impact of child's behaviour and emotions** | Parent survey | Control and intervention group surveys completed online via REDCap. | Baseline and 4 months | SDQ Impact Supplement[a] measuring the impact of the child's behaviour and emotions on their functioning and the family. |
| **Secondary outcome: Parent mental health and distress** | Parent Survey | Control and intervention group surveys completed online via REDCap. | Baseline and 4 months | Depression, Anxiety and Stress Scale[a] (DASS-21) |
| **Secondary outcome: Health service use** | Parent Survey | Control and intervention group surveys completed online via REDCap. | Baseline and 4 months | Survey of services used in the past 4 months for the child's emotional or behavioural problems, including number of times accessed, distance travelled to service, and any out of pocket costs ($AUD). |

[a]Validated measure

- Family impact, degree of burden and impairment from the child's behaviour and emotions as measured by parent-reported SDQ Impact Supplement [40].

- Parent mental health and distress as measured by the Depression, Anxiety and Stress Scale (DASS-21), a 21-item, validated measure of adult emotions [41, 42].

- Health services use for the child's emotional or behavioural problems, and out of pocket costs in Australian dollars, as reported by parents. Health service use is measured at 4-months post randomisation only.

Demographic characteristics of the user will be recorded by parent survey at baseline.

The feasibility of these secondary outcomes will also be assessed by reviewing how many parents complete the secondary outcome measures. This will be used to inform the design of a later trial powered to assess the effectiveness of the FindWays website.

## Participant timelines

Participants will be recruited over an estimated three-month period in mid 2022 using a rolling recruitment strategy. After participants have consented and completed their baseline measures, they will be randomised and followed over 4 months. At the end of the 4 months, participants will fill out a second survey. Those in the intervention group will be invited to participate in a semi-structured interview. A visual summary of the timeline and assessment points can be found in Fig 1.

## Sample size

There is no consistent advice regarding the number of participants required to assess acceptability and feasibility. Typically, between 20–100 participants are recruited into similar trials assessing acceptability, feasibility and usability [28, 30, 32]. For this trial, we anticipate recruiting up to 60 participants, within a three month time frame. As a pilot RCT, the trial is not

| | TRIAL PERIOD | | | |
|---|---|---|---|---|
| | Enrolment | Allocation to intervention | Post allocation | Close-out |
| TIME POINT** | $-t_1$ | $t_0$ | $t_1$ | $t_{+4\ months}$ |
| ENROLMENT: | | | | |
| Eligibility screen | X | | | |
| Informed consent | X | | | |
| Allocation to intervention | | X | | |
| INTERVENTIONS: | | | | |
| Intervention group | | ◆——————————◆ | | |
| Control group | | ◆——————————◆ | | |
| ASSESSMENTS: | | | | |
| Child behaviour, parent mental health survey | X | | | X |
| Demographics survey | X | | | |
| DHI usage as measured by Google Analytics | | | X (continuous) | X |
| Acceptability survey (FindWays group only) | | | | X |
| Qualitative interview (Optional consent from FindWays group only) | | | | X (optional) |
| Health service use survey | | | | X |

**Fig 1. SPIRIT schedule of enrolment, interventions, and assessments.**

powered to evaluated effectiveness of the intervention compared to control for parent, child or health service-related outcomes.

## Recruitment

The study design, including recruitment and allocation, is summarised in Fig 2.

**Screening eligible referrals.** Paediatricians will screen incoming referrals for eligibility. Paediatricians will flag referrals as potentially eligible if they appear to meet inclusion criteria and do not meet exclusion criteria. These potentially eligible participants will then be contacted via telephone by the administration staff at each clinic. The administration staff will request consent to pass on the parent's contact details to DP to hear more about the study. Those who do not consent to passing on their contact details will have their non-consent recorded without any identifying information, to help ascertain recruitment rates.

**Phone parents direct.**   For those parents who wish to hear more about the study, the administration staff will take and record verbal consent from the parent to pass on their contact details (parent name, email address, phone number and postcode) to DP. At this time, the clinic will also post the parents a copy of the participant information statement. DP will then contact the parents to tell them more about the study, check they received the participant information statement and answer any questions.

## Assignment of interventions

**Sequence generation.**   A statistician, not involved in the analysis of the trial results, will prepare the randomisation schedule. The randomisation schedule will be created using computer-generated random numbers before the first participant has been recruited, in a one-to-one ratio. The participant cohort will be stratified by child age (two-six year olds and seven-twelve year olds) and clinic (clinic one, clinic two and clinic three). Within each stratum, permuted block randomisation will be used to ensure balance between the intervention and control group. A randomly generated sequence of block sizes containing two, four, or six participants will be used. This will help prevent any predictability when randomising participants to intervention or control [43].

**Allocation concealment.**   The schedule will be held by the independent statistician, and allocation will not be revealed prematurely to DP. Because of these procedures, the research team will be unable to predict which group participants will be allocated to.

**Implementation.**   When a participant has consented to participate, and completed their baseline measures, DP will contact the independent statistician so they can reveal the participant's randomisation status. Both DP and the participant will become aware of which trial group they were allocated to after randomisation.

**Blinding.**   The participant and researcher will not be blind to their intervention status because it is impossible to blind a novel website intervention to participants. The administration staff and paediatricians will not be notified by the researchers of the intervention status of the participants.

## Withdrawing from the trial

Participants are free to withdraw from the trial at any time upon request. If known, a brief reason will be recorded on the participant database. All parents in the intervention group who withdraw will be asked if they are still willing to participate in the final interview to understand their experience using the platform. Withdrawing from the trial will not affect their access to standard treatment or their relationship their paediatrician. If they withdraw, they will be asked to no longer access the FindWays website. For participants that do withdraw, we will

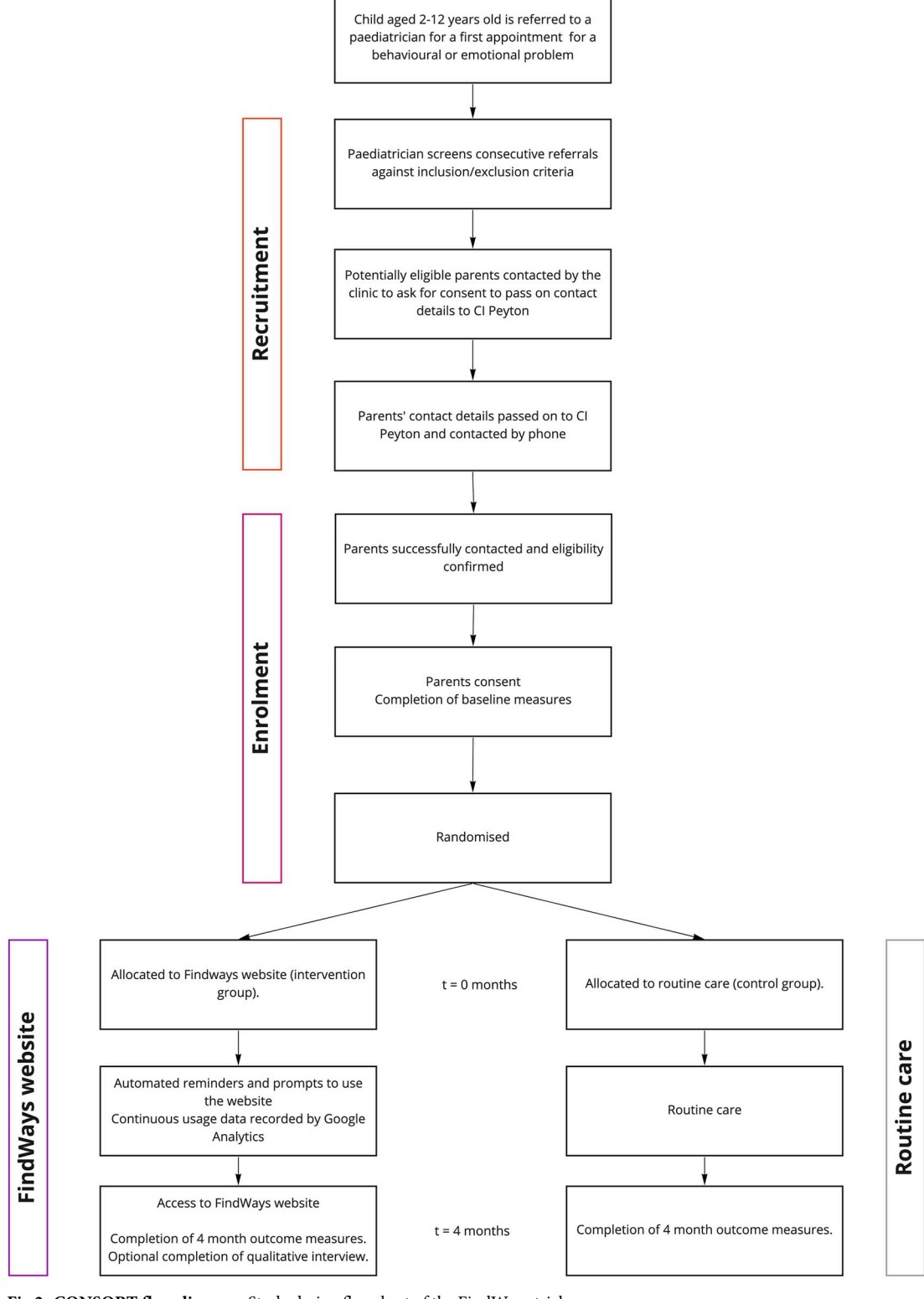

**Fig 2. CONSORT flow diagram.** Study design flowchart of the FindWays trial.

report on whether they continued to access the FindWays website using their unique link by reviewing Google Analytics data.

## Ancillary and post-trial care

Participants in the intervention group will have ongoing access to the FindWays website after the trial is over. Participants in the control group will also have access to the FindWays website after completion of their final survey four months after randomisation. Participants in both the intervention and control group will be free to utilise any treatments (other than the intervention) available to them.

## Data collection and management

Data will be collected and entered using electronic data collection forms which will be completed by the participant. Website usage data will be collected by Google Analytics and manually transferred across to the relevant participant record on REDCap, by DP. Google Analytics collects website usage data. For this trial, each participant will be sent a unique link with an individual Google Analytics tracking code. Google Analytics will then be able to show individual usage data for each participant.

Participants in the intervention group will be asked if they are willing to participate in an optional telephone semi-structured interview. DP will conduct all the phone interviews as per the interview guide. The interview will last approximately 20–30 minutes and is designed to better understand the parent's experience using the platform, enquire about any adverse events (including safety issues), barriers to use, or ways the website could be improved. Participant interviews will be recorded via dictaphone and all audio recordings will be transcribed verbatim.

Hard copy data will be stored in a locked cabinet in a secure location, accessible to the research team only. Electronic data will be securely stored in MCRI's REDCap database system and in files stored in MCRI's network file servers, which are backed up nightly.

REDCap is hosted on MCRI infrastructure and is subject to the same security and backup regimen as other systems (e.g., the network file servers).

## Statistical analysis

The baseline characteristics of the intervention and control groups will be summarised and presented separately. The primary outcome data will be summarised and presented as percentages.

For secondary outcomes linear regression and logistic regression will be conducted to estimate mean differences (and 95% confidence intervals) for continuous outcomes, and odds ratios (and 95% confidence intervals) for binary outcomes, respectively, between trial groups. Analyses will be adjusted for baseline scores of the outcome measure. This pilot RCT will be reported in accordance with the CONSORT e-health statement and we intend to complete a multiple imputation intention-to-treat analysis at the level of the child [44].

Qualitative interviews will be analysed using an inductive content analysis approach. This approach employs three main phases: i) open coding; ii) creating categories by cross referencing, and grouping the data; and, iii) abstraction [45, 46]. This analysis will be conducted using NVivo software. Data will be coded by DP and discussed with supervising authors on a regular basis. The coding framework will be reviewed by each of the co-authors. The analysis will describe key themes and events in parents' experience using the FindWays website.

## Ethical considerations

This protocol, the informed consent document and any subsequent amendments was reviewed and approved prior to commencing the research. Ethics approval granted by The Royal Children's Hospital Human Research Ethics Committee. HREC/75854/RCHM-2021 on 22/4/2022. A letter of protocol approval by HREC was obtained prior to the commencement of the trial, as well as approval for other trial documents requiring HREC review. Amendments will be communicated to investigators, ISRCTN and publishing journals.

The protocol is registered on ISRCTN (ISRCTN64605513). As this is a pilot study, a data monitoring committee is not necessary and interim analysis will not be conducted. Safety is being measured by recording individual level quantitative participant data on mental health and health service-related outcomes, as well as through qualitative interviews offered to all participants in the intervention group.

DP will conduct the informed consent discussion and will check that the parent comprehends the information provided. DP will answer any questions about the trial.

The parent will be invited to provide verbal consent. Consent will be voluntary and free from coercion. Optional consent will be obtained by DP from participants in the intervention group, who choose to participate in the semi-structured interview four months after randomisation.

Participant confidentiality is strictly held in trust by the participating investigators, research staff, and the sponsor. This confidentiality is extended to cover the health information of the participants and will not be released without written permission of the participant, except as necessary for monitoring by HREC or regulatory agencies, or as required by law.

## Dissemination policy and results

Upon completion of the study, results will be disseminated via four methods: i) Publication of results in a peer-reviewed journal; ii) presentation of results at conferences; iii) presentation of results to local paediatricians in Geelong; and iv) plain language summary of results distributed to parents agreeing to receive pilot results.

Up to January 2022, 32 participants have been recruited. Participants are expected to be enrolled until February 2022. The final outcome measures are expected to be collected in June 2023, with publication of results expected in December 2023.

The authors do not intend to use professional writers for any part of the publication.

## Discussion

This pilot RCT will determine whether a co-designed DHI, designed to help parents find strategies and services to help their child, is feasible and acceptable among parents referred to a paediatrician for their child's behavioural or emotional problems. This will be the first evaluation of its kind of a DHI for parents targeting mental health problems in young children. It will also be the first randomised trial to measure the effects of a self-directed website targeting parents on uptake of mental health services.

Limitations of this protocol include parents being recruited from a single regional city, and the study not sufficiently powered to evaluate for efficacy of the website. However, following the pilot RCT, we plan to conduct a fully powered RCT to determine the effectiveness and cost-effectiveness of this intervention in reducing the need for paediatrician services and improving child and parent outcomes The recruitment and retention rates in this pilot will inform the later RCT design, to allow for adequate time to recruit a fully powered sample. Further, the first author will conduct the qualitative interviews and the participants are aware of

their involvement in the design of the research. This could possibly result in a bias towards positive feedback.

In the long term, the FindWays website is potentially impactful because: i) it may help families improve their child's behaviour and emotions, without needing to see a paediatrician, or ii) they can find and engage with available evidence-based services earlier.

Mental health workers may also find the website useful for identifying strategies, programs and providers that are known to help children with a particular issue and available in their community. These mental health workers could include wellbeing officers at schools, general practitioners, well child/family nurses, and even paediatricians.

If this DHI is a feasible and acceptable way of linking some families to the right treatments, without the need to consult a paediatrician, it could increase the efficiency of the health system. With fewer families on waitlists, children with more complex or severe problems may be seen sooner. The overall capacity of the health system may be increased as we expect families to access group parenting programs and scalable online programs, where there is capacity to see more children than through individual face-to-face appointments with a specialist.

## Supporting information

**S1 File. SPIRIT checklist.**
(PDF)

**S2 File. Participant information statement.**
(PDF)

**S3 File. Full study protocol.**
(DOCX)

## Acknowledgments

The authors would like to acknowledge the co-design participants for their valuable input.

The Victorian Government's Operational Infrastructure Support Program support research at the Murdoch Children's Research Institute.

## Author Contributions

**Conceptualization:** Daniel Peyton, Greg Wadley, Naomi Hackworth, Harriet Hiscock.

**Formal analysis:** Daniel Peyton.

**Investigation:** Daniel Peyton.

**Methodology:** Daniel Peyton, Greg Wadley, Naomi Hackworth, Anneke Grobler, Harriet Hiscock.

**Project administration:** Daniel Peyton.

**Software:** Daniel Peyton, Greg Wadley.

**Supervision:** Greg Wadley, Naomi Hackworth, Anneke Grobler, Harriet Hiscock.

**Writing – original draft:** Daniel Peyton.

**Writing – review & editing:** Daniel Peyton, Greg Wadley, Naomi Hackworth, Anneke Grobler, Harriet Hiscock.

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
