## [Decision Letter · Decision Letter 0]

18 Nov 2022

PONE-D-22-22500

A co-designed website (FindWays) to improve mental health literacy of parents of children with mental health problems: protocol for a pilot randomised controlled trial.

PLOS ONE

Dear Dr. Peyton,

Thank you for submitting your manuscript to PLOS ONE. After careful consideration, we feel that it has merit but does not fully meet PLOS ONE’s publication criteria as it currently stands. Therefore, we invite you to submit a revised version of the manuscript that addresses the points raised during the review process.

We look forward to receiving your revised manuscript.

Kind regards,

Alison L. Calear

Academic Editor

PLOS ONE

Journal Requirements:

4. We note you have included a table to which you do not refer in the text of your manuscript. Please ensure that you refer to Table 1 in your text; if accepted, production will need this reference to link the reader to the Table.

5. We note that the original protocol file you uploaded contains a confidentiality notice indicating that the protocol may not be shared publicly or be published. Please note, however, that the PLOS Editorial Policy requires that the original protocol be published alongside your manuscript in the event of acceptance. Please note that should your paper be accepted, all content including the protocol will be published under the Creative Commons Attribution (CC BY) 4.0 license, which means that it will be freely available online, and any third party is permitted to access, download, copy, distribute, and use these materials in any way, even commercially, with proper attribution.

Therefore, we ask that you please seek permission from the study sponsor or body imposing the restriction on sharing this document to publish this protocol under CC BY 4.0 if your work is accepted. We kindly ask that you upload a formal statement signed by an institutional representative clarifying whether you will be able to comply with this policy. Additionally, please upload a clean copy of the protocol with the confidentiality notice (and any copyrighted institutional logos or signatures) removed.

Reviewers' comments:

Reviewer's Responses to Questions

**Comments to the Author**

1. Does the manuscript provide a valid rationale for the proposed study, with clearly identified and justified research questions?

Reviewer #1: Yes

Reviewer #2: Yes

Reviewer #3: Yes

2. Is the protocol technically sound and planned in a manner that will lead to a meaningful outcome and allow testing the stated hypotheses?

Reviewer #1: Partly

Reviewer #2: Yes

Reviewer #3: Yes

3. Is the methodology feasible and described in sufficient detail to allow the work to be replicable?

Reviewer #1: Yes

Reviewer #2: Yes

Reviewer #3: Yes

4. Have the authors described where all data underlying the findings will be made available when the study is complete?

Reviewer #1: Yes

Reviewer #2: Yes

Reviewer #3: No

5. Is the manuscript presented in an intelligible fashion and written in standard English?

Reviewer #1: Yes

Reviewer #2: Yes

Reviewer #3: Yes

6. Review Comments to the Author

You may also provide optional suggestions and comments to authors that they might find helpful in planning their study.

Reviewer #1: This is a well written and conceived protocol for a pilot evaluation of a website for parents of children referred to a paediatrician for help with their child’s emotional and behavioural problems. The need for such a resource is well articulated and the evaluation includes quantitative and qualitative outcomes to comprehensively understand its feasibility and acceptability. I have only minor comments that should be addressed to improve the manuscript.

• In the background section, where does the 1 in 3 statistic for mental health problems in children come from? The Lawrence 2015 reference is ~1 in 7.

• Rather than anecdotal estimates of wait times for paediatricians, is there no data available on this to cite, particularly post-covid? This paragraph would be strengthened with a supporting reference, given its central importance to the intervention rationale (p5)

• Could the authors explain why children with a diagnosis of autism spectrum disorder are excluded, given the comorbidity between autism and mental health problems? Would these parents be excluded from using the website in practice? If so, is there an alternative resource that the researchers can direct these parents to?

• Add a reference to Table 1 in the text.

• Figures 1 and 2 are very blurry and hard to read

• How will parents who withdraw from the study not have access to the Findways website? Do parents have to log in to use it?

• It would be better if the 4-month follow-up participant interviews were conducted by a team member who was not involved in the design of the website, as participants may not feel comfortable disclosing negative feedback to the interviewer, and your findings may be biased towards only positive outcomes.

• Please provide more information on how you will conduct an intention to treat analysis when you have missing data. Will you use a method of imputation or mixed models? A completers analysis is not consistent with ITT.

Reviewer #2: This protocol focuses on the acceptability and feasibility of a new intervention ‘FindWays’ which has been developed to support parents while they wait to see a paediatrician for their child’s emotional or behavioural problem. The rationale is clearly stated in the introduction and the method concisely describes the protocol and allows testing of the aims.

A few minor considerations for the authors to address:

Introduction

1. The manuscript is focused on children with emotional and behavioural problems, barriers to seeking help, and supports available. Paragraph 2 of the introduction is focused on evidence-based treatments to improve child emotional and behavioural problems, however reference 9 (Merry et al., 2012) is focused on an adolescent help-seeking intervention. Considering this reference is aimed at a different age cohort (12-19 year olds) to that of interest in this publication, I would suggest removing this reference and replacing it with something more age appropriate (if needed).

Methods

2. Line 296 states “Secondary outcomes measured by parent-reported surveys at 4-months post randomisation”. These outcomes measures were also collected at baseline (as indicated in Table 1). Please include “at baseline and” to make it clearer for the reader.

SPIRIT Checklist

3. The checklist has been completed fully, however some page number/paragraph numbers are not accurate. For example, 2a is on page 19, 16a is on pages 15-16, 17a is on page 16, 24 is on page 18, 27 is on page 19. Please update this checklist once all modifications have been incorporated.

Reviewer #3: This paper describes the protocol of a pilot RCT, that plans to access feasibility of digital platform as an intervention to improve health literacy of parents with children mental health problems. The paper correctly focuses on feasibility as it should be done for a pilot RCT and not on efficacy of the proposed intervention. As a result, proposed sample size is based on convenience and not based on statistical significance. I suggest authors to discuss little bit about the “primary outcome(s)” as well as other non-primary (secondary, exploratory) outcome(s) and how the feasibility of those will be assessed. The authors may also point out shortcomings of the study if any and how the pilot RCT pilot will help them in guiding future fully powered RCT design. Overall, a very nicely written paper with minor changes needed.

7. PLOS authors have the option to publish the peer review history of their article (what does this mean?). If published, this will include your full peer review and any attached files.

Reviewer #1: **Yes: **Amy Morgan

Reviewer #2: No

Reviewer #3: No

---

## [Author Response · Author response to Decision Letter 0]

26 Feb 2023

Editor comments

We have checked the manuscript aligns with the PLOS ONE style templates. 

Thank you for pointing this out. We have updated the funding information in the metadata attachment to ensure it aligns with the financial disclosure section and updated the grant numbers across both sections. 

We have now included the ethics statement in the manuscript Methods section. 

4. We note you have included a table to which you do not refer in the text of your manuscript. Please ensure that you refer to Table 1 in your text; if accepted, production will need this reference to link the reader to the Table. 

Thank you for identifying this error. We now refer to Table 1 in the text on page 12 line 277. 

“See Table 1 for a summary of the outcome measures and data collection time points.”

5. We note that the original protocol file you uploaded contains a confidentiality notice indicating that the protocol may not be shared publicly or be published. Please note, however, that the PLOS Editorial Policy requires that the original protocol be published alongside your manuscript in the event of acceptance. Please note that should your paper be accepted, all content including the protocol will be published under the Creative Commons Attribution (CC BY) 4.0 license, which means that it will be freely available online, and any third party is permitted to access, download, copy, distribute, and use these materials in any way, even commercially, with proper attribution.

Therefore, we ask that you please seek permission from the study sponsor or body imposing the restriction on sharing this document to publish this protocol under CC BY 4.0 if your work is accepted. We kindly ask that you upload a formal statement signed by an institutional representative clarifying whether you will be able to comply with this policy. Additionally, please upload a clean copy of the protocol with the confidentiality notice (and any copyrighted institutional logos or signatures) removed.

We have now included a new version of the protocol (version 1.4.2), approved by the Royal Children’s Hospital Human Research Ethics Committee that has removed reference to this confidentiality statement. 

We have double checked each reference, and to the best of our knowledge, we have not cited any retracted manuscripts. 

Reviewer comment

Reviewer 1

Minor comments 

In the background section, where does the 1 in 3 statistic for mental health problems in children come from? The Lawrence 2015 reference is ~1 in 7. 

Thank you for clarifying this error in frequency of mental health problems. As stated, mental health disorders affect one in 7 Australian children. We have also included a statement of the increased prevalence of mental health problems requiring treatment, as perceived by parents, which affect 26.8% of children from Lawrence 2015 publication on page 4 line 72. 

“Diagnosed mental health disorders, such as disruptive behaviour, anxiety and mood problems, affect one in seven Australian children [1]. However, mental health problems, which can still impact the child but may not reach diagnostic criteria for a mental health disorder, These more common behavioural and emotional problems needing treatment may affect around one in four children. 

Rather than anecdotal estimates of wait times for paediatricians, is there no data available on this to cite, particularly post-covid? This paragraph would be strengthened with a supporting reference, given its central importance to the intervention rationale (p5) 

Thank you for this suggestion. Unfortunately, there are no new published data on waiting times to see a paediatrician for a mental health problem since COVID. The only reference quantifying the waitlist to see a paediatrician has already been included - Mulraney et al 2021.

Could the authors explain why children with a diagnosis of autism spectrum disorder are excluded, given the comorbidity between autism and mental health problems? Would these parents be excluded from using the website in practice? If so, is there an alternative resource that the researchers can direct these parents to? 

Autism spectrum disorder was not included within the website for a few reasons. Firstly, the FindWays website is designed for those waiting to see a paediatrician, and who would not have yet received a formal diagnosis. Children with an autism diagnosis generally have already seen a paediatrician (as a paediatrician or child psychiatrist is required for diagnosis). Secondly, families of children with autism, or who might be waiting for an autism diagnosis, already have access to websites to address their specific needs, including Amaze website and the A-list Hub. Finally, there are often specific treatment needs for autistic children that don’t lend themselves to have this information co-located with mental health information for the general population (hence why there are already existing websites such as Amaze). This might include difficulties distinguishing autistic traits from anxiety traits, that may require different therapeutic options. 

If, in the course of the research, there are families wanting more resources who are ineligible to take part because of an existing autism spectrum disorder diagnosis, then we would recommend speaking to their current therapist or paediatrician, or perusing the Amaze website. 

Add a reference to Table 1 in the text. 

We have now included a reference to Table 1 in the text.

Figures 1 and 2 are very blurry and hard to read 

The figures are blurry when viewed within the PDF manuscript. However, when they are downloaded by clicking the link at the top of the figure, they are clear and readable. 

How will parents who withdraw from the study not have access to the Findways website? Do parents have to log in to use it? 

There is no log in to the website. If a parent wishes to withdraw from the study, they will be asked to not access the website. We can report on whether they user continue to access the FindWays website through their unique link. To make this clearer we have amended the text to now read on page 17 line 389:

“If they withdraw, they will be asked to no longer access the FindWays website. For participants that do withdraw, we will report on whether they continued to access the FindWays website using their unique link by reviewing Google Analytics data.”

It would be better if the 4-month follow-up participant interviews were conducted by a team member who was not involved in the design of the website, as participants may not feel comfortable disclosing negative feedback to the interviewer, and your findings may be biased towards only positive outcomes. 

We agree that having an alternative team member conduct the interviews would potentially reduce biased feedback. However, the research team are limited by available resources as this study is part of a PhD. We now acknowledge that this is a limitation of the study on page 21 line 487. 

“Further, the first author will conduct the qualitative interviews and the participants are aware of their involvement in the design of the research. This could possibly result in a bias towards positive feedback.”

Please provide more information on how you will conduct an intention to treat analysis when you have missing data. Will you use a method of imputation or mixed models? A completers analysis is not consistent with ITT. 

Thank you for this feedback. We intend to complete multiple imputation intention-to-treat analysis. We have clarified this plan in the manuscript on page 19 line 428. 

“This pilot RCT will be reported in accordance with the CONSORT e-health statement and we intend to complete a multiple imputation intention-to-treat analysis at the level of the child [44].”

Reviewer comment 

Reviewer 2

Paragraph 2 of the introduction is focused on evidence-based treatments to improve child emotional and behavioural problems, however reference 9 (Merry et al., 2012) is focused on an adolescent help-seeking intervention. Considering this reference is aimed at a different age cohort (12-19 year olds) to that of interest in this publication, I would suggest removing this reference and replacing it with something more age appropriate (if needed). 

Thank you for this comment. We have now removed this reference. We will not replace it as there is no published evidence for digital health interventions improving depression symptoms in children under 12 years old. 

2. Line 296 states “Secondary outcomes measured by parent-reported surveys at 4-months post randomisation”. These outcomes measures were also collected at baseline (as indicated in Table 1). Please include “at baseline and” to make it clearer for the reader. 

We have updated this sentence to make it clearer. We have also added a short statement about the health services use survey on page 14 line 301 and on page 14 line 311, which is measured at 4 months only. 

“Secondary outcomes measured by parent-reported surveys at baseline and 4-months post randomisation include…”

And

“Health service use is measured once at 4-months post randomisation.”

3. The checklist has been completed fully, however some page number/paragraph numbers are not accurate. For example, 2a is on page 19, 16a is on pages 15-16, 17a is on page 16, 24 is on page 18, 27 is on page 19. Please update this checklist once all modifications have been incorporated. 

Thank you for pointing out this error. We have checked and updated the number/paragraph numbers on the checklist. 

Reviewer comment 

Reviewer 3

I suggest authors to discuss little bit about the “primary outcome(s)” as well as other non-primary (secondary, exploratory) outcome(s) and how the feasibility of those will be assessed. 

We have interpreted this comment as an opportunity to describe in more detail how to measure feasibility, both in the context of the primary and secondary outcomes. 

For our primary outcomes, we have provided more detail on measuring feasibility using our primary outcomes on page 7 line 147. 

“Despite the lack of a singular definition of feasibility, they typically involve measures of recruitment rates, attrition rates, evidence of harm, user satisfaction and DHI usage [29,30] and will be assessed in the following ways:”

We have also included a brief description of how we will assess the feasibility of the secondary outcomes on page 14 line 314.

“The feasibility of these secondary outcomes will also be assessed by reviewing how many parents complete the secondary outcome measures. This will be used to inform the design of a later trial powered to assess the effectiveness of the FindWays website.”

The authors may also point out shortcomings of the study if any and how the pilot RCT pilot will help them in guiding future fully powered RCT design. Overall, a very nicely written paper with minor changes needed.

Thank you for this feedback. We have included the following sentence when referring to the fully powered RCT on page 21 line 485. 

“The recruitment and retention rates in this pilot will inform the later RCT design, to allow for adequate time to recruit a fully powered sample.”

---

## [Editor Report · Decision Letter 1]

6 Mar 2023

A co-designed website (FindWays) to improve mental health literacy of parents of children with mental health problems: protocol for a pilot randomised controlled trial.

PONE-D-22-22500R1

Dear Dr. Peyton,

We’re pleased to inform you that your manuscript has been judged scientifically suitable for publication and will be formally accepted for publication once it meets all outstanding technical requirements.

Kind regards,

Alison L. Calear

Academic Editor

PLOS ONE
---

## [Editor Report · Acceptance letter]

10 Mar 2023

PONE-D-22-22500R1 

A co-designed website (FindWays) to improve mental health literacy of parents of children with mental health problems: protocol for a pilot randomised controlled trial. 

Dear Dr. Peyton:

I'm pleased to inform you that your manuscript has been deemed suitable for publication in PLOS ONE. Congratulations! Your manuscript is now with our production department. 

Kind regards, 

on behalf of

Dr. Alison L. Calear 

Academic Editor

PLOS ONE